



# The impact of sampling strategy on the cloud droplet number concentration estimated from satellite data

Edward Gryspeerdt[1,2], Daniel T. McCoy[3], Ewan Crosbie[4,5], Richard H. Moore[4], Graeme J. Nott[6], David Painemal[4,5], Jennifer Small-Griswold[7], Armin Sorooshian[8], and Luke Ziemba[4]

[1]Space and Atmospheric Physics Group, Imperial College London, UK
[2]Grantham Institute for Climate Change and the Environment, Imperial College London, UK
[3]Department of Atmospheric Science, University of Wyoming, Laramie, WY, USA
[4]NASA Langley Research Center, Science Directorate, Hampton VA, USA
[5]Science Systems and Applications Inc., Hampton, VA, USA
[6]FAAM, Cranfield, UK
[7]University of Hawai'i at Mānoa, Honolulu, HI, USA
[8]Department of Chemical and Environmental Engineering, University of Arizona, Tucson, AZ, USA

**Correspondence:** Edward Gryspeerdt (e.gryspeerdt@imperial.ac.uk)

**Abstract.** Cloud droplet number concentration ($N_d$) is of central importance to observation-based estimates of aerosol indirect effects, being used to quantify both the cloud sensitivity to aerosol and the base state of the cloud. However, the derivation of $N_d$ from satellite data depends on a number of assumptions about the cloud and the accuracy of the retrievals of the cloud properties from which it is derived, making it prone to systematic biases.

A number of sampling strategies have been proposed to address these biases by selecting the most accurate $N_d$ retrievals in the satellite data. This work compares the impact of these strategies on the accuracy of the satellite retrieved $N_d$, using a selection of insitu measurements. In stratocumulus regions, the MODIS $N_d$ retrieval is able to achieve a high precision ($r^2$ of 0.5-0.8). This is lower in other cloud regimes, but can be increased by appropriate sampling choices. Although the $N_d$ sampling can have significant effects on the $N_d$ climatology, it produces only a 20% variation in the implied radiative forcing

from aerosol-cloud interactions, with the choice of aerosol proxy driving the overall uncertainty. The results are summarised into recommendations for using MODIS $N_d$ products and appropriate sampling.

## 1   Introduction

The droplet number concentration ($N_d$) is a key property of clouds. As the first moment of the droplet size distribution, the $N_d$ is important for setting cloud and precipitation process rates (e.g. Khairoutdinov and Kogan, 2000) as well as cloud

radiative properties (George and Wood, 2010; Painemal, 2018). It is closely related to the aerosol environment and the in-cloud updraught (Twomey, 1959), as well as being affected by precipitation processes (Wood, 2012) and entrainment (Baker et al., 1980). With this important role for cloud properties, $N_d$ has been used to evaluate the performance of global climate models (Mulcahy et al., 2018; McCoy et al., 2020).



Variations in $N_d$ are also a primary method for observational characterisations of aerosol effects on clouds (e.g. Quaas et al., 2006). An increase in available cloud condensation nuclei (CCN) will typically produce an increase in $N_d$, which can result in changes in droplet size and cloud reflectivity (Twomey, 1974), modifications to precipitation processes (Albrecht, 1989), intensification of convection (Williams et al., 2002) as well as increases in evaporation and potential cloud desiccation (Wang

et al., 2003; Ackerman et al., 2004). This has made aerosol relationships with $N_d$ the target of a large number of observational studies (e.g. Quaas et al., 2006, 2008; Ghan et al., 2016; Gryspeerdt et al., 2017; McCoy et al., 2017; Hasekamp et al., subm). With a central role in aerosol-cloud interactions, $N_d$ relationships with other cloud properties, particularly cloud fraction (CF; Gryspeerdt et al., 2016) and liquid water path (LWP; Han et al., 2002), have also been used to quantify cloud adjustments due to aerosols.

Assessments of the effective radiative forcing due to aerosol-cloud interactions (ERFaci) rely heavily on these observation-based estimates of aerosol-cloud interactions (Boucher et al., 2013; Bellouin et al., 2020) and these estimates in turn rely on accurate observations of aerosol-$N_d$ and $N_d$-cloud relationships. Reliable satellite and remotely sensed observations of $N_d$ are therefore essential for reducing uncertainties in the anthropogenic impact on clouds and the climate.

There are a number of methods for retrieving cloud droplet size and $N_d$ from space (Boers et al., 2006; Zeng et al., 2014;

Austin and Stephens, 2001; Hu et al., 2021), but the majority of previous studies make use of the cloud droplet number calculated from a bispectral retrieval of the cloud optical depth ($\tau_c$) and effective radius ($r_e$; Nakajima and King, 1990), assuming an adiabatic cloud (Boers et al., 2006; Quaas et al., 2006). Previous studies in stratocumulus regions have found a good agreement between the satellite and insitu $N_d$ (Painemal and Zuidema, 2011), but this retrieval depends on assumptions with varying applicability (Grosvenor et al., 2018b). To improve our knowledge of the $N_d$ across the globe, a number of

sampling strategies have been applied in recent work to select more reliable retrievals (Quaas et al., 2006; Grosvenor et al., 2018b; Bennartz and Rausch, 2017; Zhu et al., 2018), based on the characteristics of the retrieval and the observed liquid clouds.

Each of these sampling strategies are based on an understanding of cloud physics and the character and reliability of satellite retrievals, such that it is not immediately clear which is most suitable for selecting valid $N_d$ retrievals. In addition, as $N_d$

products are used for a variety of different tasks, different sampling methods may be more appropriate for each. Removing low optical depth clouds may limit the $N_d$ retrieval to more accurate cases, but may produce a biased climatology and estimates of the ERFaci by neglecting a large fraction of the cloud population (Leahy et al., 2012). This work examines these sampling strategies and how the choices made impact the accuracy of the $N_d$ retrieval when compared to insitu data, the representative-ness of the $N_d$ climatology and the impact of these choices on the implied aerosol-cloud radiative forcing.





## 2 Methods

### 2.1 $N_d$ from satellite

$N_d$ is rarely retrieved directly, but is estimated from the cloud optical depth ($\tau_c$) and effective radius ($r_e$). Assuming an adiabatic cloud (no precipitation or mixing with its environment), the $N_d$ is derived from the retrieved properties ($\tau_c$, $r_e$) following (Brenguier et al., 2000; Quaas et al., 2006; Boers et al., 2006).

$$N_d = \frac{1}{2\pi k} \sqrt{\frac{5}{Q\rho_w}} (f_{ad}c_{ad})^{1/2} \tau_c^{1/2} r_e^{-5/2} \tag{1}$$

where the density of water $\rho_w$ and the scattering efficiency $Q$ (=2) are assumed constant. $k = (r_v/r_e)^3$, where $r_v$ is the droplet mean volume radius, depends on the droplet size spectrum. Although $k$ has been observed to vary in insitu studies (Martin et al., 1994) and it may vary under particularly extreme aerosol environments (Noone et al., 2000), this work uses a constant value of 0.8, following Painemal and Zuidema (2011) and Grosvenor and Wood (2014) .

The condensation rate $c_{ad}$ is a function of temperature and pressure. The pressure dependence is weak, but the temperature dependence can produce a 50% variation in the $N_d$. To account for this variation, the $N_d$ is calculated using the linear $N_d$ temperature correction from Gryspeerdt et al. (2016), using the cloud top temperature (a suitable assumption if the cloud layers are thin; Grosvenor and Wood, 2014).

The sub-adiabatic factor ($f_{ad}$) in Eq. 1 represents the reduction in the condensation rate due to mixing with sub-saturated environmental air. However, a full accounting for sub-adiabaticity also requires a potential change in the droplet size distribution (except under extreme inhomogeneous mixing), which modifies the $k$ parameter. Previous work has suggested that there might be a cancellation between these two effects (Painemal and Zuidema, 2011). Observational studies have found a range of values for the adiabatic factor from 0.74 (Kang et al., 2021), 0.8 (Braun et al., 2018), 0.88 (Painemal et al., 2017) and 0.9 (Painemal and Zuidema, 2013). In this work a constant factor of 0.8 is used, noting that this may be responsible for an offset in the retrieved $N_d$.

### 2.2 Satellite sampling

Two of the major uncertainties in the $N_d$ retrieval are the cloud adiabaticity assumption and the accuracy of the cloud retrievals used to derive the $N_d$. This work examines sampling strategies to minimise these uncertainties in the MODIS collection 6.1 cloud optical properties retrieval (MOD06_L2) dataset for both Aqua and Terra (Platnick et al., 2017). This is a bispectral retrieval (Nakajima and King, 1990), with known uncertainties in broken cloud situations and where there are large variations in the effective radius (Zhang and Platnick, 2011). The $N_d$ sampling methods in this work (Tab. 1) aim to reduce these uncertainties through sampling retrievals with higher confidence.

Only liquid water clouds can be considered here, so our analysis is restricted to cases with a valid optical properties retrieval and a retrieved liquid water phase. As a baseline strategy, this sampling method is referred to as ("All") throughout this work. Unless otherwise noted, the $r_e$ and $\tau_c$ values come from the standard MODIS 2.1 μm retrieval.





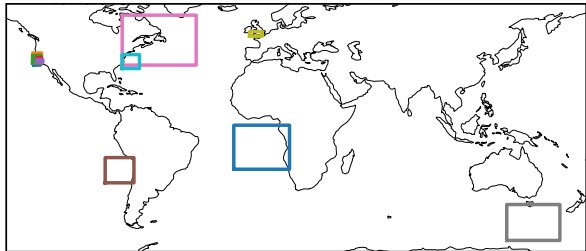

**Figure 1.** Locations of the campaigns used in this work. Colours are shown in Fig. 2

With a high uncertainty in $r_e$ retrievals at a low $\tau_c$ and a degeneracy in the retrievals for a low $r_e$, Quaas et al. (2006) suggested the exclusion of cases with a $\tau_c$ or $r_e$ less than 4 (or 4 µm) when calculating the $N_d$. This sampling is hence called "Q06".

Grosvenor and Wood (2014) demonstrated the uncertainties at high solar zenith and satellite viewing angles, where cloud 3D

effects and multiple scattering generate uncertainties, particularly in the $r_e$. The non-linear nature of these retrievals can also bias retrievals in broken cloud and inhomogeneous scenes (Zhang and Platnick, 2011). Recognising these issues, Grosvenor et al. (2018b) makes several recommendations to avoid these problematic retrievals. Following these, cases with a solar zenith angle greater than 55°, a satellite viewing zenith angle >41.1° and a cloud mask SPI (sub-pixel inhomogeneity index, the standard deviation relative to the mean of the 250m radiances; Liang et al., 2009) greater than 30% are excluded. This is in

addition to the Q06 sampling. This sampling strategy is named "G18".

These sampling strategies focus primarily on the properties of the retrievals. However, the cloud adiabaticity plays an important role in the $N_d$ retrieval. The final two methods make attempts to address this. Bennartz and Rausch (2017) propose a method for locating adiabatic pixels by comparing the $r_e$ at different wavelengths. The $r_e$ retrieved using the 3.7 µm band is typically located closer to the cloud top than the 2.1 µm and 1.6 µm $r_e$ retrievals, due to the wavelength dependence of water

absorption (Platnick, 2000). For an adiabatic cloud, the $r_e$ at 3.7 µm is therefore expected to be larger than the shorter wavelengths (and $r_e$ at 2.1 µm $> r_e$ at 1.6 µm), although other factors including retrieval biases can also impact these relationships (Zhang and Platnick, 2011). Only pixels satisfying these inequalities (known as $r_e$-stacking) are included in this sampling method. As it is applied ontop of G18, it is more stringent than the sampling proposed in Bennartz and Rausch (2017), but is named "BR17" due to the importance of the $r_e$-stacking criterion.

Finally, Zhu et al. (2018) suggest that the adiabatic fraction can be maximised by only using data from cloud "cores" - the 10% highest $\tau_c$ values in 100 by 100 km regions. This is applied on top of the G18 sampling, and called "Z18". As with BR17, this is more stringent than Zhu et al. (2018), due to the additional filters inherited from G18.

These sampling strategies are all applied at 1 km resolution (L2/pixel level). The L2 retrievals are aggregated to daily means at a 1° by 1° resolution for aerosol susceptibility calculations.





| All | Liquid phase |
|-----|-----|
| | Single layer |
| | Cloud top temperature $> 268K$ |
| Q06 | All and |
| | $r_e > 4\,\mu m$ |
| | $\tau_c > 4$ |
| G18 | Q06 and |
| | 5km CF $> 0.9$ |
| | Solar Zenith $< 55°$ |
| | Sat. Zenith $< 41.4°$ |
| | Cloud SPI $< 30$ |
| BR17 | G18 and |
| | $r_e\,(3.7\,\mu m) > r_e\,(2.1\,\mu m) > r_e\,(1.6\,\mu m)$ |
| Z18 | G18 and |
| | $\tau_c$ in top 10% |

**Table 1.** Summary of sampling methods.

## 2.3 Aircraft data selection

To assess these sampling methods, satellite retrievals are compared to aircraft measurements of $N_d$. A selection of aircraft data is used to provide a variety of different cloud and meteorological conditions, including marine stratocumulus (a key region for the radiative forcing from aerosol-cloud interactions), mid-latitude stormtracks and the Southern Ocean (Fig. 1).

Stratocumulus data comes from the CIRPAS Twin Otter data in Sorooshian et al. (2018), including data from the E-PEACE, FASE, MACAWS, MASE1 and MASE2 campaigns. These campaigns took place over the northeast Pacific near the coast (Fig. 1). These campaigns had a consistent use of the CASF (the forward scattering component of the cloud, aerosol and precipitation spectrometer) and a large number of intersections with the MODIS instrument. For these campaigns the liquid water content (LWC) comes from the PVM-100A probe on the Twin Otter. Data from the NCAR C-130 during VOCALS-

REx (VAMOS Ocean-Cloud-Atmosphere-Lands Study - Regional Experiment Wood et al., 2011) provides measurements of a different stratocumulus region. The C-130 used a cloud droplet probe (CDP) to measure the droplet size spectrum. Data from the phase doppler interferometer (PDI) onboard the P-3 during the ORACLES (ObseRvations of Aerosols above CLouds and their intEractionS Redemann et al., 2021) are used to provide $N_d$ measurements of the Namibian stratocumulus deck. Only data from 2016 and 2018 are used, due to issues with the PDI in 2017.

Four other flight campaigns are used to investigate the $N_d$ retrieval in a broader range of clouds, often in more challenging conditions. Data for North Atlantic boundary layer clouds comes from the North Atlantic Aerosol and Marine Ecosystem (NAAMES) campaign (Behrenfeld et al., 2019). A CDP was used to measure the droplet size distribution during a three year period (2015-2017). Data from the ACTIVATE (Aerosol Cloud meTeorology Interactions oVer the western ATlantic





Experiment; Sorooshian et al., 2019) includes $N_d$ data from a CDP during 2020, aimed primarily at shallow liquid clouds (cumulus and winter postfrontal stratocumulus) off the Eastern Coast of the USA. The SOCRATES (Southern Ocean Clouds, Radiation, Aerosol Transport Experimental Study; McFarquhar et al., 2021), aimed at Southern Ocean clouds, provides CDP observations of $N_d$ in a challenging, often mixed phase environment. Finally, COPE (Convective Precipitation Experiment

Leon et al., 2016) used a CDP to measure $N_d$ in convective environments. For the COPE campaign, LWC data comes from the Johnson-Williams instrument; for all other campaigns using a CDP, the LWC is calculated from the CDP size distribution.

For each flight campaign, 1Hz data is used. For the CDP instruments, the total particle number (2-50 μm) is used. For the campaigns using CASF and PDI data, bins are selected (with a linear interpolation for partial bins) to produce a $N_d$ representative of the range 2-30 μm (the exact values have little effect on the results presented in this work).

## 2.4 Insitu data sampling

As the aim of this work is to evaluate the satellite sampling strategies and products, extensive filtering on the aircraft data is not performed, relying on the satellite to select cases where there are valid $N_d$ retrievals (as is required for an global product). In particular, no attempt is made to select the $N_d$ at cloud top. While the $N_d$ retrieval uses cloud top $r_e$, it is based on the assumption that $N_d$ is constant throughout the cloud depth. This assumption is valid on average for VOCALS-REx (Painemal

and Zuidema, 2011), SOCRATES (Kang et al., 2021) and NAAMES (Painemal et al., 2021), but may not be for a non-adiabatic cloud. A satellite retrieval has to be able to identify these situations.

The LWC-$N_d$ relationship in Fig. 2 shows a very strong relationship at low LWC values, likely due to inhomogeneous mixing reducing the $N_d$ and LWC at cloud edges (Baker et al., 1980). To ensure that the insitu $N_d$ measurements are representative of the whole cloud, rather than a mixing region close to a cloud edge, a uniform minimum LWC of 0.1 g m$^{-3}$ is used, discarding

aircraft $N_d$ measurements below this when comparing to the satellite retrievals.

The aircraft data is aggregated and compared to MODIS data at a pixel level (1 km by 1 km at nadir). For each MODIS pixel, all the 1Hz aircraft data within that pixel (that satisfy the sampling criteria) are averaged together. A pixel must have more than 2 aircraft points (2 seconds) of data to be included in this analysis. To minimise errors from cloud motion and cloud development, a co-incidence time between the satellite and aircraft data of less than 15 minutes is required.

## 2.5 Aerosol data

To assess the impact of $N_d$ sampling techniques implied radiative forcing from aerosol-cloud interactions (RFaci), the susceptibility of $N_d$ to aerosol ($\beta$) variations is calculated (Feingold, 2003).

$$\beta_{N_d} = \frac{d\ln N_d}{d\ln A} \tag{2}$$

where $A$ is an aerosol proxy. Three aerosol proxies are used in this work, with all $\beta$s calculated at 1° by 1° resolution. The

aerosol optical depth (AOD) is a simple proxy used in previous work (e.g. Quaas et al., 2008), but that underestimates the aerosol impact on clouds (Gryspeerdt et al., 2017). The aerosol index (AI), defined as the AOD multiplied by the Angström





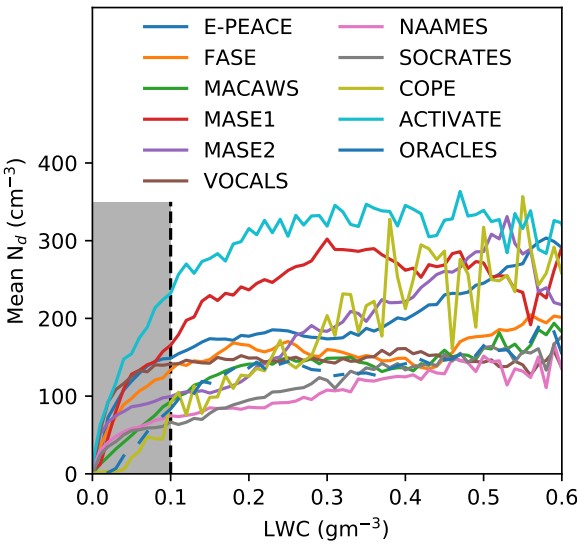

**Figure 2.** Average aircraft $N_d$ as a function of LWC. Aircraft data with an LWC less than $0.1\,\mathrm{gm}^{-3}$ is excluded from this analysis.

exponent (Nakajima et al., 2001) is able to diagnose the RFaci to within 20%, provided accurate retrievals of AI and $N_d$ (Gryspeerdt et al., 2017). Following Hasekamp et al. (subm), AI retrievals less than 0.1 are discarded due to their high uncertainty. Both AOD and AI are from the daily mean MODIS collection 6.1 1° by 1° product (MYD08_D3). The AOD is the combined dark target (Levy et al., 2013) and deep blue (Sayer et al., 2014) product, while the AI is calculated from the AOD-Angström exponent joint histograms over ocean only. Reanalysis aerosol products are also a potential aerosol proxy, correlating well to $N_d$ in a variety of environments (McCoy et al., 2017). The MERRA2 900hPa $SO_4$ concentration is also used as an aerosol proxy, as in McCoy et al. (2017).

To estimate the contribution of sensitivity variations to the implied RFaci, the RFaci is calculated as

$$\mathrm{RFaci} = F^{\downarrow} f_c \frac{\alpha_c (1 - \alpha_c)}{3} \beta_{N_d} \Delta A \tag{3}$$

Where $F^{\downarrow}$ is the CERES downwelling flux, $f_c$ is the MODIS liquid cloud fraction and $\alpha_c$ is the cloud albedo, derived from the MODIS cloud optical depth. These estimates are calculated at a 1° by 1° resolution.

## 3 Results

### 3.1 Satellite-insitu comparison (pixel level)

Given the large number of assumptions and uncertainties in the $N_d$ retrieval, the agreement between MODIS and insitu $N_d$ is surprisingly good (Fig. 3). Coefficients of determination ($r^2$) for the stratocumulus campaigns are in the range 0.5 to 0.8




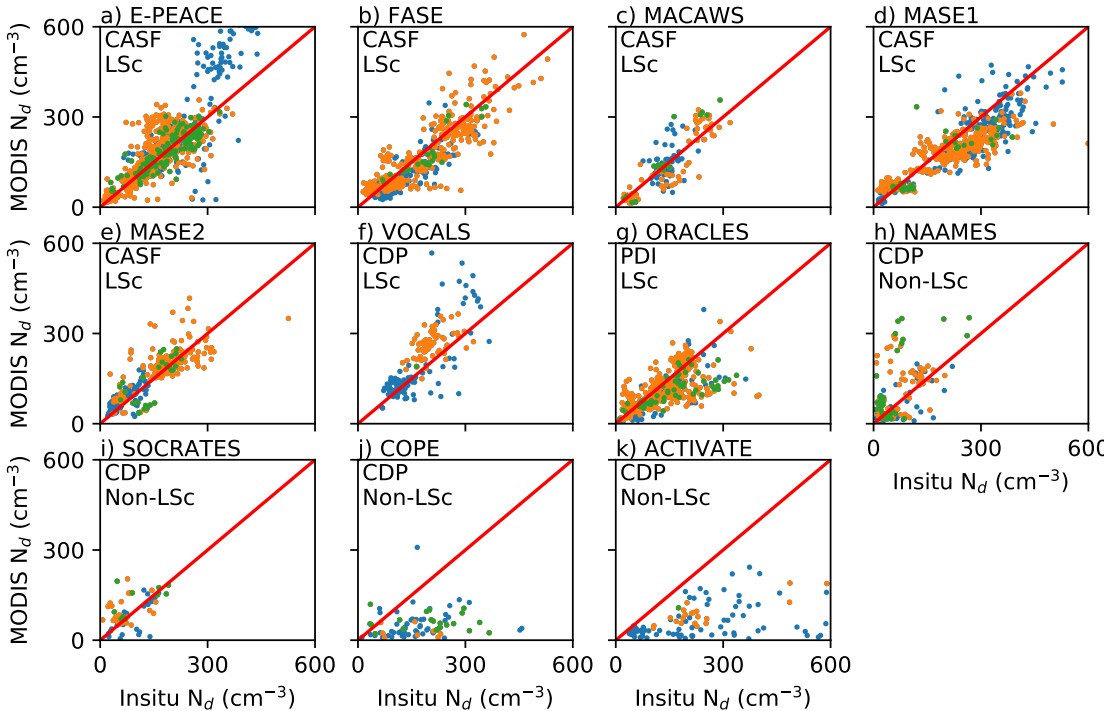

**Figure 3.** Comparison between MODIS and insitu $N_d$ at a pixel level, for aircraft data within 15 minutes of an MODIS (Aqua or Terra) overpass. Blue is data that doesn't satisfy G18, orange dots are G18 sampling and green are Z18. The instrument used in each campaign and the main cloud type (LSc - Liquid stratocumulus) is shown in the top left of each subplot.

(Tab. 2). For the more challenging situations, the coefficient of determination is lower (in the range 0.25 to 0.5), but still shows skill at retrieving $N_d$.

Even for the least stringent filtering ("All"), $r^2$ values remain high for the stratocumulus campaigns (as in Painemal and Zuidema, 2011). This agreement holds even for some of the large $N_d$ values ($500cm^{-3}$) seen in E-PEACE (Fig. 3a) and FASE

5 (Fig. 3b), even though these pixels are removed by the G18 and Z18 sampling strategies as potentially biased.

The retrievals for most of the stratocumulus campaigns have high $r^2$ values (Tab. 2) and close alignment to the 1:1 line (Fig. 3). However, In some of the more challenging situations, particularly NAAMES and SOCRATES, MODIS can overestimate the insitu values (Fig. 3h-k), sometimes by more the $100cm^{-3}$. Even the more stringent sampling strategies of G18 and Z18 are unable to identify these pixels as biased, suggesting that further filtering techniques are be required to provide accurate

10 $N_d$ values under these circumstances.

All the sampling strategies fail to accurately characterise the $N_d$ from COPE. Convective clouds are a uniquely challenging environment for the $N_d$ retrieval, with strong mixing limiting potential adiabatic locations (Eytan et al., 2021). Not only does this limit the applicability of Eq. (1), the extremely heterogeneous clouds limit the accuracy of the MODIS retrievals (Zhang and Platnick, 2011) and large variations in $N_d$ increase representation errors for the aircraft data. The comparisons





|  | All | Q06 | G18 | BR17 | Z18 |
|---|---|---|---|---|---|
| E-PEACE | 0.67 | 0.71 | 0.59 | 0.78 | 0.69 |
| FASE | 0.77 | 0.77 | 0.77 | 0.79 | 0.83 |
| MACAWS | 0.74 | 0.79 | 0.82 | 0.82 | 0.91 |
| MASE1 | 0.72 | 0.69 | 0.66 | 0.46 | 0.69 |
| MASE2 | 0.70 | 0.69 | 0.51 | 0.61 | 0.51 |
| VOCALS | 0.62 | 0.63 | 0.27 | 0.45 | - |
| ORACLES | 0.46 | 0.45 | 0.45 | 0.63 | 0.51 |
| NAAMES | 0.24 | 0.22 | 0.23 | 0.17 | 0.46 |
| SOCRATES | 0.29 | 0.26 | 0.29 | 0.45 | 0.32 |
| COPE | 0.02 | 0.06 | 0.00 | - | 0.01 |
| ACTIVATE | 0.02 | 0.38 | 0.67 | - | - |
| Average | 0.48 | 0.51 | 0.48 | 0.57 | 0.55 |
| All | 0.36 | 0.44 | 0.46 | 0.71 | 0.42 |

**Table 2.** Coefficient of determination ($r^2$) for MODIS-Insitu comparisons for the 2.1 μm retrieval. "-" indicates too few points to calculate a correlation. The "Average" row is the average $r^2$ across the campaigns and the "All" row is the $r^2$ for all the valid data points across all campaigns.

with ACTIVATE are slightly better, especially for the more restrictive sampling strategies. Even so, MODIS typically produces underestimates of the $N_d$ when compared to the insitu data. This is expected in broken cloud and inhomogeneous scenes, which lead to overestimates in the $r_e$ (Zhang and Platnick, 2011) and corresponding underestimates in the $N_d$.

Considering all the available pixel-level matches between MODIS and the insitu data, BR17 produces the strongest overall correlation, with an $r^2$ of 0.71 (Tab. 2). Both Q06 and G18 are improvements on using all data, with only a 10 and 25% reduction in the data volume respectively. In comparison, BR17 discards almost 63% of available liquid cloud pixels. Interestingly, while Z18 often produces high correlations to the insitu data, the overall $r^2$ (0.42) is lower than any other sampling strategy. This is partly due to it preferentially selecting sub-adiabatic convective retrievals in the more convective campaigns (e.g. COPE), as it selects the highest optical-depth cases. Although the correlation in a single campaign can be high, the bias varies between campaigns and so produces a worse correlation overall.

## 3.2 Other sampling choices

### 3.2.1 Should I use a minimum cloud fraction?

G18 introduces filtering by the 5 km CF, ensuring the retrieval is more than 2 km from a cloud edge. While this reduces the impact of cloud inhomogeneities, some studies have required a high 1° by 1° liquid CF to further reduce the impact of this uncertainty (e.g. Grosvenor et al., 2018a). This can remove broken cloud scenes where retrieval uncertainties can be higher.

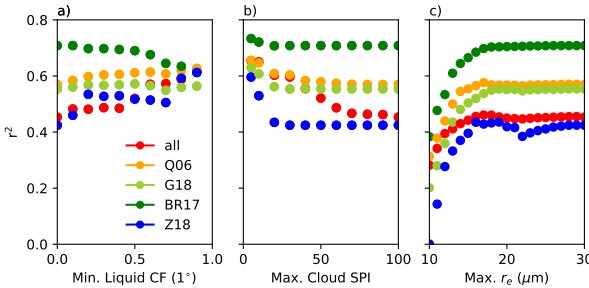

**Figure 4.** The impact of filtering by (a) large scale liquid cloud fraction, (b) pixel-level cloud SPI and (c) the maximum permitted $r_e$ on the total $r^2$ for each sampling strategy.

Specifying a minimum large-scale liquid CF has a relatively small impact on the $r^2$ (Fig. 4a), with a gradual increase in the total $r^2$ as the minimum cloud fraction increases for the majority of sampling strategies. There is a corresponding decrease in the data volume, only around 50% of investigated pixels have a total liquid CF > 90%, but it would improve the accuracy of the remaining retrievals if that was the only consideration.

The Z18 sampling shows a slightly larger increase in $r^2$ as the minimum CF increases, becoming the highest accuracy strategy for high liquid CF (Fig. 4a). This is likely due to the cloud core assumption of Z18 being most valid for closed-cell stratocumulus cases. This suggests that while the Z18 sampling might be less suited to broken-cloud cases, it could be preferred in high liquid CF environments.

### 3.2.2    Which SPI threshold should I use?

G18 also introduces a cloud mask SPI threshold, which aims to exclude pixels with sub-pixel variation in cloud properties. Gryspeerdt et al. (2019) used a maximum value of 30%, finding that further limiting this value made little difference to their results. However, for the pixel-level MODIS-insitu comparison (Fig. 3), limiting the SPI further produces a measurable increase in the accuracy of the MODIS $N_d$ retrieval (Fig. 4b), particularly for the Z18 sampling strategy.

Using a maximum SPI of 5% reduces the available data with the "all" strategy by 45%. This is only a 29% reduction for the

Z18 strategy (where SPI is already limited to a maximum of 30%; Tab. 1). If a higher accuracy is required, a lower SPI limit can help achieve this. A very strict SPI limit significantly reduces the accuracy difference between the sampling strategies and may be a more data efficient way to achieve close-to BR17 accuracy levels than $r_e$-stacking (Fig. 4b).

### 3.2.3    Should I use a maximum $r_e$?

Large cloud top $r_e$ has been proposed as an indicator of warm rain (Rosenfeld and Ulbrich, 2003). As a precipitating cloud is

non-adiabatic, this creates a systematic bias as a function of $r_e$. Restricting the $N_d$ calculation to a maximum $r_e$ may increase the overall accuracy of the sampled $N_d$.



For all the sampling strategies, setting a very low maximum $r_e$ (<15 μm) results in a reduction in the accuracy of the $N_d$ retrieval by removing most of the data being studied (Fig. 4c). A very high maximum $r_e$ recovers the values from Tab. 2. For most of the sampling strategies, there is a slight increase in accuracy between these two limits, with a maximum correlation between the MODIS and insitu $N_d$ for a maximum $r_e$ of around 15 μm. The increase in accuracy is stronger for the Z18 strategy, possibly as it targets retrievals in cloud cores where precipitation is more likely. In these situations, an extra precipitation filter would have the clearest effect. In contrast, a maximum $r_e$ has no impact on the BR17 filtering, as the $r_e$ stacking is already designed to filter out precipitating cases.

### 3.2.4 Which wavelength should I use?

|  | $r^2$ (All) | | | $r^2$ (Non-Sc) | | |
|---|---|---|---|---|---|---|
|  | 2.1 μm | 1.6 μm | 3.7 μm | 2.1 μm | 1.6 μm | 3.7 μm |
| All | 0.45 | 0.42 | 0.42 | 0.19 | 0.13 | 0.23 |
| Q06 | 0.57 | 0.54 | 0.51 | 0.32 | 0.21 | 0.37 |
| G18 | 0.55 | 0.51 | 0.49 | 0.38 | 0.21 | 0.41 |
| BR17 | 0.71 | 0.62 | 0.70 | 0.58 | 0.31 | 0.57 |
| Z18 | 0.42 | 0.43 | 0.39 | 0.25 | 0.19 | 0.26 |

**Table 3.** The impact of $r_e$ wavelength on the total $r^2$. The second set of values are for only the non-stratocumulus campaigns.

The standard MODIS $r_e$ retrieval uses the 2.1 μm band. In broken cloud and inhomogeneous conditions, the 3.7 μm $r_e$ retrieval is expected to produce more accurate $r_e$ retrievals (Zhang and Platnick, 2011; Painemal and Zuidema, 2013). For ideal clouds, the 3.7 μm retrieval retrieves $r_e$ closer to the cloud top and the 1.6 μm retrieval deeper into the cloud. With potential compensating errors, it is not clear which wavelength retrieval gives the best $N_d$.

The agreement between the MODIS and insitu $N_d$ values depends on the absorbing wavelength used in the joint $\tau_c$-$r_e$ retrieval (Tab. 3). When considering all the data together, the 1.6 μm retrieval has a higher $r^2$ and the 3.7 μm a slightly lower $r^2$ for all the sampling strategies, other than BR17 (where the effective radius stacking criterion imposes a strict relationship between $r_e$ at different wavelengths).

Considering all the campaigns together hides the behaviour in more challenging situations. In non-stratocumulus situations, the 1.6 μm $N_d$ retrieval does not perform as well as the standard (2.1 μm) retrieval, whereas the 3.7 μm retrieval performs better than the standard (Tab. 3, right three columns). The variation in non-stratocumulus campaigns is consistent with inhomogeneity generated biases in $r_e$ retrievals, where the 3.7 μm retrieval performs better in broken cloud environments (Zhang and Platnick, 2011). In these broken-cloud regions, the 3.7 μm retrieval could be preferred.

For the stratocumulus campaigns, the difference in $r^2$ as a function of wavelength is much smaller, but is slightly higher for the 1.6 μm retrieval. This might be an indicator of cloud top entrainment reducing the $r_e$, with the 1.6 μm retrieval being more accurate as it is focused further inside the cloud. If the cloud top entrainment is particularly extreme, the $r_e$ stacking would not be satisfied, so BR17 limits the impact of this effect. Although the difference between the different wavelength retrievals





in stratocumulus situations is small, it may affect correlations between $N_d$ and other cloud properties, particularly the liquid water path (LWP). Cloud-top mixing has been proposed as contributing to the negative $N_d$-LWP relationship (Gryspeerdt et al., 2019).

### 3.3 Should I correct for penetration depth biases?

The derivation of Eq. 1 assumes the $r_e$ is from the cloud top, but satellite retrievals provide $r_e$ at a distance below the cloud top, based on the photon penetration depth (Platnick, 2000). This low bias in the $r_e$ is hypothesised to lead to a high bias in $N_d$, particularly for thin clouds (Grosvenor et al., 2018a).

Applying the Grosvenor et al. (2018a) correction for penetration depth results in a reduced high $N_d$ bias at high $N_d$ for the VOCALS and E-PEACE campaigns (not shown). For the other campaigns, there is either little change or a decrease in $N_d$

retrieval accuracy. This may be due to compensating biases in the $N_d$ retrieval and the Q06 sampling removing cases with low optical depths where this penetration depth bias is strongest. Although this correction is not applied in this work, as the quality of $N_d$ retrievals improves, the penetration depth bias may play a more important role in the overall $N_d$ error budget.

### 3.4 Satellite-insitu comparison (1° by 1°)

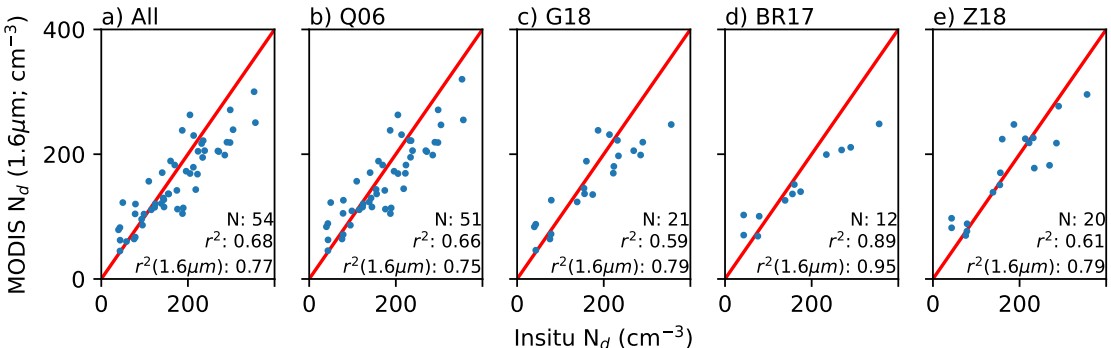

**Figure 5.** Comparison of 1° by 1° mean insitu and MODIS $N_d$. Requires at leave 300 insitu measurements and 1500 valid MODIS retrievals in a gridbox. Each scatterplot also shows the number of points and the $r^2$ value, along with the $r^2$ value for the 1.6 μm retrieval in the bottom right.

Many studies using MODIS $N_d$ derived data do so at 1° by 1° resolution. Although insitu data has difficulty representing

such a large region, it is instructive to make a simple comparison between MODIS and insitu data at this resolution (see also McCoy et al., 2020). It is not possible to collect aircraft data to perfectly characterise entire gridbox this size. To increase the representation of the data for each gridbox, 300 seconds of in-cloud aircraft data and more than 1500 MODIS pixels are required for each 1° by 1° gridbox. Only 150 MODIS pixels are required for the Z18 mask, as it makes an explicit aim to select fewer but more representative MODIS pixels.





The correlations between the insitu and MODIS data are high (Fig. 5), with $r^2$ values above 0.7 even when considering all available liquid pixels. This is considerably higher than the pixel-level correlations in Tab. 2. The correlations increase for the more restrictive sampling methods, although there is a corresponding decrease in the number of valid gridboxes. The $r^2$ reaches over 0.9 for BR17, increasing still further when using the 1.6 μm retrieval (Fig. 5). Although the strategy requiring

a large coverage of MODIS and insitu data biases this comparison toward high CF, stratocumulus regimes where the $N_d$ assumptions are more likely to hold, this comparison gives increasing confidence that the MODIS $N_d$ retrieval is capable of accurately retrieving the $N_d$ at a pixel level and over larger regions.

## 4    Applications

### 4.1    Representing the $N_d$ climatology

A key requirement for an $N_d$ retrieval is the ability to represent the $N_d$ climatology, especially if it is being used to constrain model simulations (Mulcahy et al., 2018; McCoy et al., 2020). This is already conceptually difficult, as a model maintains an $N_d$ even in situations with a very low LWC where a satellite or aircraft is unable to measure an $N_d$, requiring the use of a satellite simulator.

Fig. 6 shows how well each of the satellite sampling strategies represents the climatology of insitu $N_d$ data for all the

potential locations in each campaign. For each sampling strategy, the number of remaining datapoints is shown along the x-axis.

In general, the satellite sampling strategies all do a good job representing the climatology, particularly in stratocumulus regions (as expected following their agreement in this regime, see Fig. 7). However, for NAAMES, both BR17 and Z18 appear to slightly overestimate the mean $N_d$ for the campaign. This may also be the case for ACTIVATE, but the low number of

intersections limits our ability to draw strong conclusions. The overestimate in NAAMES appears to be due to the sampling method keeping pixels where MODIS overestimates $N_d$, whilst discarding cases with better agreement, but a lower MODIS $N_d$ (Fig. 3g).

For representing the climatology, these results suggest that G18 may be a better choice, particularly outside of stratocumulus regimes. However, the small number of satellite-aircraft comparisons in these cases limits current confidence in the accuracy

of the satellite $N_d$ climatology outside stratocumulus.

### 4.2    Satellite climatologies

The different sampling strategies for the MODIS $N_d$ produce broadly similar $N_d$ climatological patterns (Fig. 7), with higher $N_d$ values over land and in coastal regions and lower values over the remote ocean. While some previous studies have removed data over land, it is kept here as $N_d$ information over land is used for observation-based estimates of the RFaci.

The mean $N_d$ and land ocean contrast differ significantly between sampling methods. While Q06 and G18 have similar global patterns, the G18 mean is typically higher than Q06, with this increase being slightly larger over land than ocean

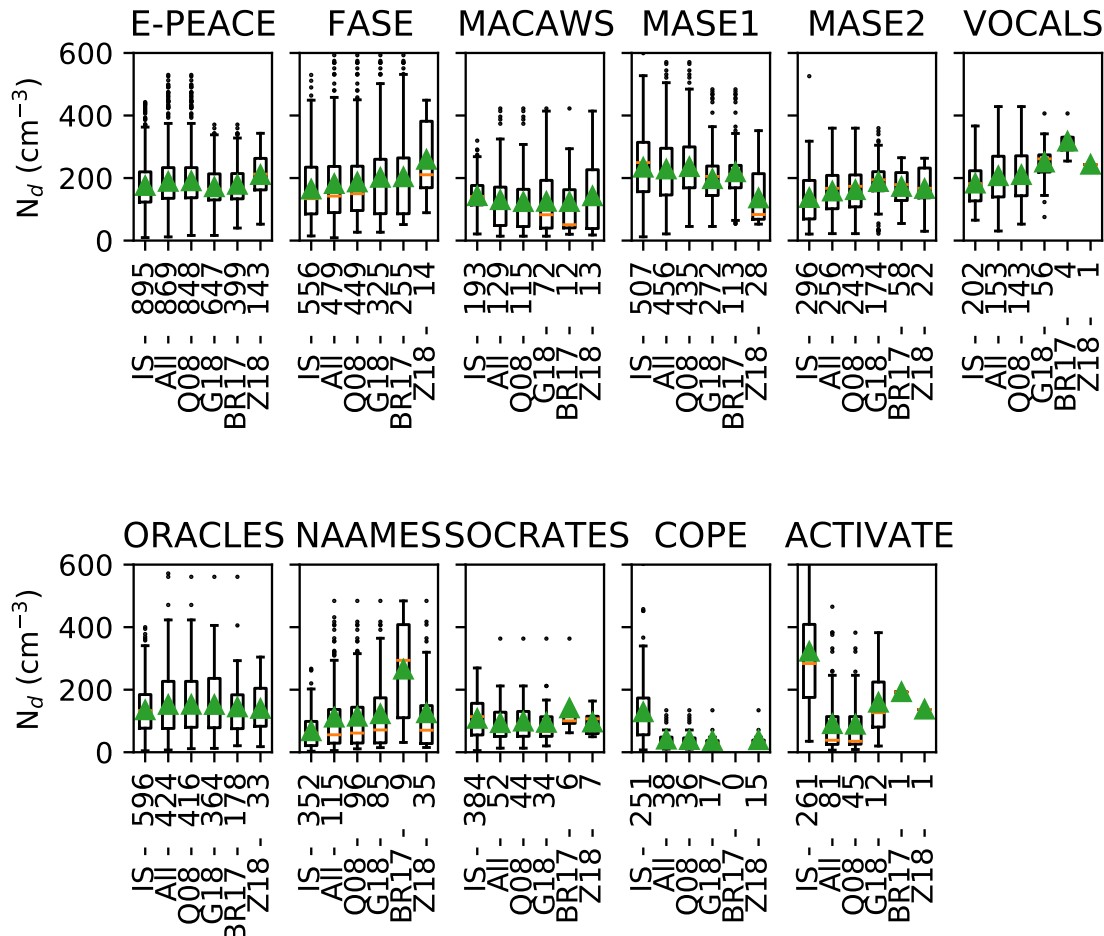

**Figure 6.** Comparison between the MODIS and insitu $N_d$ distributions for each campaign. In each subplot, the insitu distribution is the left-most boxplot, composed of all the valid insitu datapoints with a coincident (within 30 minutes) satellite view (from either Aqua or Terra), independent of whether there is a valid retrieval. Green triangles are the mean and orange lines the median. The other boxes in each subplot are the distributions of valid satellite $N_d$ retrievals for each sampling strategy that are coincident with aircraft measurements. The number of $N_d$ datapoints for each boxplot is given below the x-axis.

(Fig. 7). BR17 produces a significantly larger $N_d$ across most of the globe (particularly over land) than either Q06 or G18. Similar to BR17, the Z18 enhancement over land is also large (although smaller than BR17), but there is a smaller overall enhancement over ocean.

The difference between the sampling strategies is much smaller in stratocumulus regions, where the CF is larger. In these regions, clouds are much more likely to be adiabatic (and hence satisfy the BR17 $r_e$ stacking criterion). This means that even sampling methods that don't apply this criterion directly will satisfy it most of the time, leading to the small difference in mean





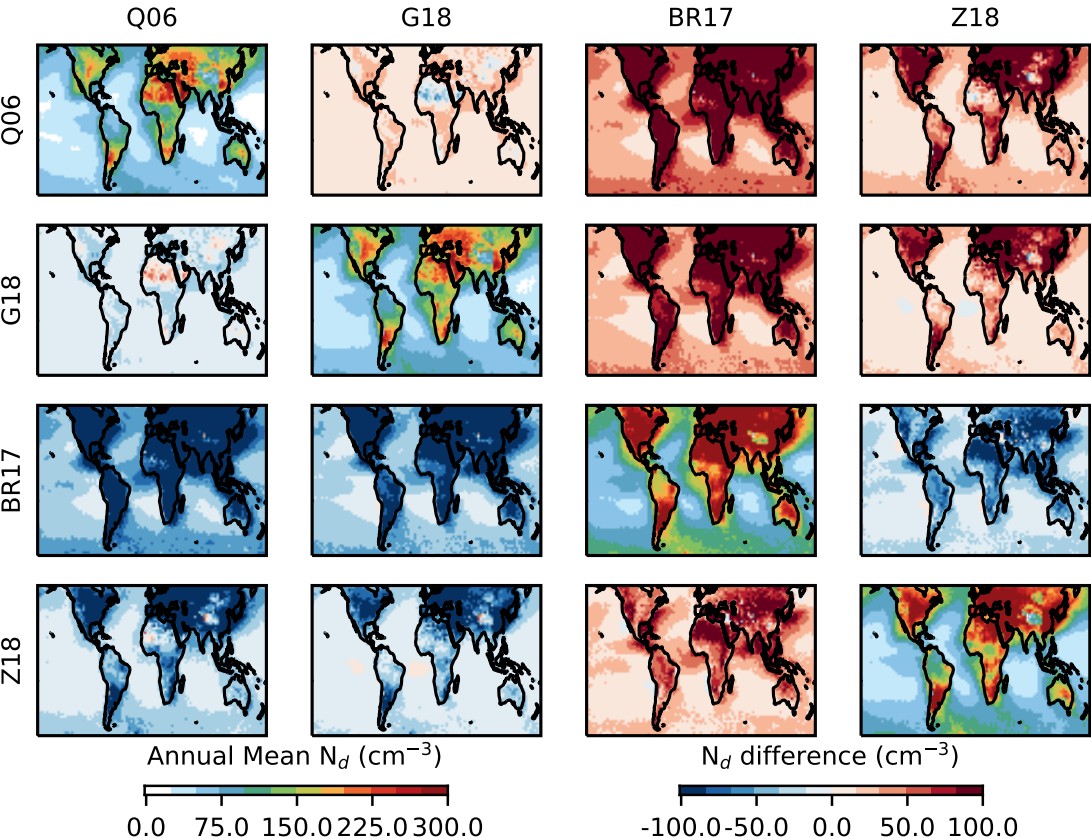

**Figure 7.** MODIS $N_d$ climatology (2011-2020) for different sampling strategy. The diagonal is the annual mean $N_d$ for each strategy, while the off-diagonal plots show the difference (e.g. the top right is Z18-Q06).

$N_d$ between the sampling methods (consistent with the results in Fig. 4a) . Over ocean, there is a significant difference in the mean $N_d$ along the Eastern coasts of North America and Asia, where liquid CF are lower and retrievals are more challenging.

### 4.3 Data coverage

The similarity between the climatologies derived from the different sampling methods hide the very different data coverage
5  (Fig. 8). With a relatively relaxed sampling criterion, Q06 has an $N_d$ retrieval in the majority of available MODIS gridboxes. This is larger than the liquid cloud fraction as only a single valid $N_d$ pixel is required to count a 1° by 1°) gridbox as "retrieved". Only regions with large ice-cloud coverage (the warm pool and over land) have a significantly lower fraction of retrievals.

With much more stringent filtering, G18 provides an $N_d$ retrieval on only around 30% of days, climbing to around 50% of days in stratocumulus regions. While much of the G18 sampling conditions are based on geometric properties, it also relies on





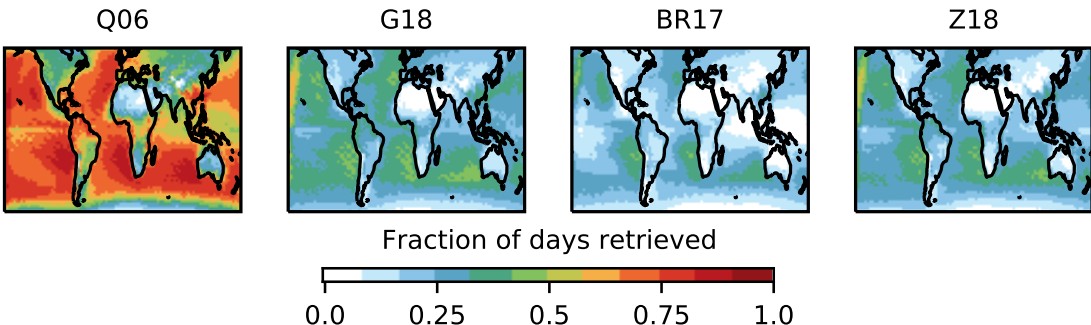

**Figure 8.** The fraction of $1°$ by $1°$ daily pixels with an $N_d$ retrieval for each sampling method.

the cloud SPI, which is typically lower in stratocumulus regions (as they are more homogeneous). This inhomogeneity criterion also contributes to the significantly reduced retrieval fraction over land.

As an even more stringent sampling strategy, BR17 has valid retrievals on an even lower fraction of days. While similar to G18 in the middle of the stratocumulus decks, the requirement for stacked $r_e$ retrievals limits the retrievals to primarily these

regions, with very few retrieved points away from stratocumulus decks. Z18 has a similar pattern to G18. As it selects just the highest 10% of $\tau_c$ within each 100km region, it can return a retrieval on any day in a gridbox where G18 has more than 10 valid retrievals, with around 25% of days having a valid $N_d$ retrieval.

### 4.4 Aerosol-cloud sensitivities and RFaci

Another major use for $N_d$ is calculating aerosol-cloud sensitivities, either for use as an emergent constraint (Quaas et al., 2009),

or for making direct estimates of the RFaci and ERFaci (e.g. Quaas et al., 2008).

As shown in Fig. 9, the sensitivity (as defined in Eq. 2) is largely unaffected by the choice of $N_d$ sampling strategy. The biggest difference appears over land, where BR17 produces a more positive sensitivity when compared to other methods.

The variations in sensitivity and its spatial pattern produce around a 20% variation in the implied RFaci (Fig. 9, lower right corners), with larger RFaci values implied when using the BR17 and Z18 strategies. The smaller impact of $N_d$ uncertainties

on the RFaci (compared to aerosol uncertainties) is expected, as $N_d$ is the independent variable in the $\beta_N$ calculation. As such, the correlation between satellite $N_d$ and true $N_d$ does not strongly affect the value of $\beta_N$ inferred from linear regression for reasonable sample sizes (e.g. larger than a few dozen).

For a simple linear regression calculation, only deviations from a linear relationship between the observed and actual $N_d$ affect the calculated $\beta_N$. Biases in $N_d$ that scale with true $N_d$ do not affect inferred $\beta_N$ because of the power law relationship

assumed in the regression. Examining the correspondence between aircraft $N_d$ and satellite $N_d$ in Fig. 3 and Fig. 5 supports a linear relationship with zero intercept, even in cases where they do not fall along the 1-1 line. Thus the $N_d$ calculation methods examined here appear to be all be of sufficient accuracy to produce accurate estimates of $\beta_N$. However, bi-variate methods for calculating $\beta_N$ (e.g. Pitkänen et al., 2016) are more sensitive to the estimates of uncertainty in the $N_d$ retrieval and would





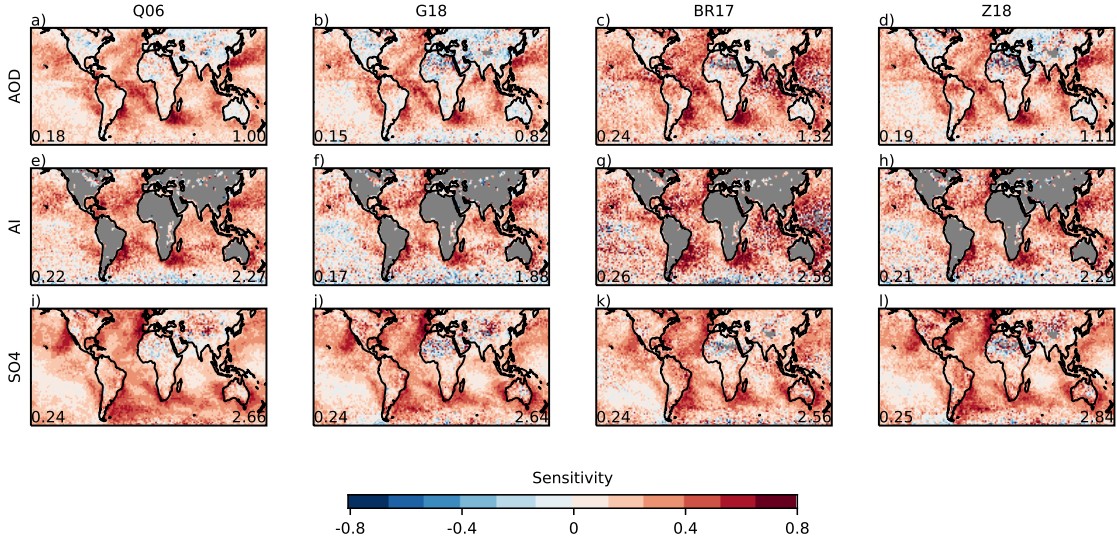

**Figure 9.** Maps of the sensitivity of $N_d$ to a selection of aerosol proxies ($\beta_N$). Each plot shows the global mean $\beta_N$ in the lower left and the ratio of the implied RFaci to that calculated using Q06 $N_d$ and AOD (a) in the lower right.

have a different error profile. In addition, as the $N_d$ is the independent variable in many calculations of cloud adjustments, the uncertainty here still has a critical role to play in the calculation of the ERFaci.

Fig. 9 demonstrates that although the aerosol proxy is still the major source of uncertainty in observation based estimates of RFaci and ERFaci, the $N_d$ sampling strategy is a non-negligible source of uncertainty because it affects the aerosol proxy

data considered and thus sampled deviations between aerosol proxy and actual CCN. It is not clear which of these sampling strategies provides the best estimate of the RFaci. Although BR17 is the most accurate at a pixel level (Tab. 2), it is based on a subset of cases which may not be representative of the overall climatology (Fig. 6). Further studies will be necessary to reduce this uncertainty.

## 5  Discussion and conclusions

The $N_d$ is an important property of clouds, both for assessing cloud models and for constraining aerosol-cloud interactions. However, its retrieval is based on a number of assumptions of varying validity. In addition, it is derived from retrievals of $\tau_c$ and $r_e$ (Eq. 1) that are themselves uncertain, inheriting potential biases from these retrievals. In recent years, a number of sampling strategies have been suggested (Tab. 1) to select cases where the assumptions are more likely to be valid and the retrievals less likely to be biased. This work investigates these assumptions and their impact on the implied radiative forcing.

At a pixel level (1km), the satellite $N_d$ (from MODIS) and insitu $N_d$ are well correlated (Fig. 3). This is especially true in stratocumulus regimes ($r^2$ in the range 0.5 to 0.8, Tab. 2), where high cloud fractions and adiabatic clouds are more common.





Even in more challenging cumulus and convective situations, the MODIS $N_d$ retrieval can provide useful information about the $N_d$, although correlations are significantly lower.

The different sampling strategies have varying strengths and weaknesses. BR17 has the strongest correlation to insitu $N_d$ across a range of aircraft campaigns, but has the lowest coverage of any of the strategies investigated (Fig. 8). While Z18 has

a lower accuracy than other strategies, it has a higher correlation to insitu $N_d$ in high CF locations. Potential improvements to the sampling strategies are demonstrated (Fig. 4), leading to a number of recommendations for the use of MODIS-derived $N_d$ products in the future.

  – A high correlation between MODIS and insitu $N_d$ is achieved even with minimal filtering. This can represent the variability in $N_d$ better than the more selective sampling methods (Fig. 6).

– BR17 appears to have the best correlation with aircraft data across a wide variety of conditions (Tab. 2), but may be biased high in broken cloud conditions (Fig. 6).

  – Z18 has a lower skill for low cloud fractions, but the accuracy increases for high cloud fractions (likely due to the validity of the assumptions used; Fig. 4).

  – The 3.7 μm retrieval is a better match to insitu data in non-stratocumulus cases, consistent with studies looking at the

effective radius retrieval. There is may be a small advantage to using the 1.6 μm retrieval in stratocumulus situations (Tab. 3) and for 1° by 1° averages (Fig. 5). However, although confidence in this is low, given the known uncertainties in the 1.6 μm $r_e$ retrieval (Zhang and Platnick, 2011). Users should be cautious if they intend to use it.

  – G18 has the closest match to the climatology (Fig. 6), although the lack of satellite-insitu comparisons in non-stratocumulus regimes reduces confidence in the climatology in these locations.

The correlation between insitu and satellite $N_d$ increases further when considering 1° by 1° averages, with $r^2$ values of 0.9 for the BR17 sampling strategy (Fig. 5). However, the uncertainty in these correlations remains high due to the small number of datapoints and the high representation errors for aircraft measurements of a 1° by 1° region.

Even with the different climatologies produced by the sampling strategies (Fig. 7), the susceptibility of $N_d$ to aerosol proxies remains remarkably similar (Fig. 9). The similarity is closest in stratocumulus regions, resulting in $N_d$ sampling generating

only a 20% variation in the implied forcing. The impact of the aerosol proxy on the estimated RFaci remains the largest uncertainty, although $N_d$ sampling produces an uncertainty of around 20%.

The apparent close agreement between MODIS and insitu $N_d$ masks a number of uncertainties. While $N_d$ measurements in stratocumulus regions agree well, there is significant diversity in the $N_d$ estimates in non-stratocumulus cases. While these are less important for the RFaci (Gryspeerdt and Stier, 2012), they may be critical for the forcing from cloud adjustments

(e.g. Koren et al., 2014) and observations of the $N_d$ in these regions are essential for constraining the magnitude of these adjustments (Gryspeerdt et al., 2016). Additionally, biases in the $N_d$ may be correlated to biases in other cloud properties (such as the LWP). Understanding and reducing these systematic biases is beyond the scope of this work, but vital to make progress in observationally constraining aerosol cloud interactions.



While significant uncertainties remain, this work has demonstrated that the MODIS $N_d$ retrieval has skill in retrieving the $N_d$ in a variety of different cloud regimes. Not only is there a close match between insitu and satellite data at a pixel level, there is a close match between the insitu and satellite $N_d$ climatologies, with a sufficient accuracy for addressing a wide range of questions in cloud and aerosol-cloud physics at global scale.

*Code availability.* Code and data available through a data repository on publication. Data from the Twin Otter campaigns are available at https://figshare.com/articles/dataset/A_Multi-Year_Data_Set_on_Aerosol-Cloud-Precipitation-Meteorology_Interactions_for_Marine_Stratocumulus_Clouds/5099983. ACTIVATE and NAAMES data are available from the Langley Atmospheric Research Centre https://www-air.larc.nasa.gov/missions.htm. VOCALS and SOCRATES data are available from UCAR at https://www.eol.ucar.edu/all-field-projects-and-deployments. COPE data is available from the NCAS British Atmospheric Data Centre (BADC) at http://catalogue.ceda.ac.uk/uuid/8440933238f72f27762005c33d2aa278.

*Competing interests.* The authors have no competing interests.

*Acknowledgements.* The authors thank the VOCALS and SOCRATES teams for their work collecting the data used in this work. EG was supported by a Royal Society University Research Fellowship (URF/R1/191602). DTM acknowledges the support of the University of Wyoming. AS was funded by ONR grant N00014-20-1-2385 and NASA grant 80NSSC19K0442 in support of the ACTIVATE Earth Venture Suborbital-3 (EVS-3) investigation, which is funded by NASA's Earth Science Division and managed through the Earth System Science
Pathfinder Program Office. Twin Otter campaigns were funded by N00014-04-1-0118, N00014-10-1-0200, N00014-11-1-0783, N00014-10-1-0811, N00014-16-1-2567, and N00014-04-1-0018.





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
