# Peer review of "The impact of sampling strategy on the cloud droplet number concentration estimated from satellite data"

_Atmospheric Measurement Techniques, 2021_

## Referee Comment (RC1)

**Review of Gryspeerdt, et al. - The impact of sampling strategy on the cloud droplet number concentration estimated from satellite data**

This paper compares MODIS droplet concentration retrievals with those from several aircraft campaigns and calculates $r^2$ values for several different sampling strategies for the MODIS pixels. The quality of the writing is good and a good number of field campaigns are included. The field was in need of such comparisons between in situ and satellite Nd data since previously this had only been done for a few campaigns. The overall scientific quality is good and lots of interesting aspects are explored. However, there are a few mistakes (concerning the values used for filtering relative to those suggested in the cited literature) and an over reliance on the $r^2$ metric, which is not always informative. Additional metrics (mean bias, RMSE, etc.) should be calculated too. The aircraft sampling method is also quite different to previous comparisons leading to more datapoints but without the ability to determine the root cause of any discrepancies (since profiles are not used). This is fine, but it would be good to explain the differences with previous work (where higher $r^2$ values were found) and to compare the effects of averaging larger numbers of pixels (e.g., 3x3, 5x5) rather than jumping to 100x100. I would also like to see uncertainties for the $r^2$ values since it is unclear whether some of the differences in $r^2$ are significant and there is likely a degree of randomness.

Here is a list giving more detail on the above and some additional comments :-

Grosvenor (2018) suggested that solar zenith angle was restricted to below 65 degrees not 55. And it was 55 degrees for the viewing angle, not 41.4. This will lead to quite a lot more data being discarded than would be the case if using the correct values. What effect does this have?

For the G18 sampling there are lots of factors being applied at once – it would be better to test them individually. Which has the largest influence? It would also be useful to test BR17 and Z18 separately from G18.

Can you provide uncertainties for the r^2 values? These would help determine the likelihood of the differences between the different sampling strategies being due to chance.

Is r^2 the best metric to use? And you should provide more details on the particular r^2 value that you are using – is it appropriate for the data populations? Some of the r^2 values don't seem to match with what might be expected in Figs. 3 and 5 (admittedly judging by eye). E.g., are the COPE r^2 values really so low? Can you double check? Adding the uncertainties would help here. It would be helpful to also plot the lines of best fit against which the r^2 values are calculated rather than just the 1:1 line. For Fig. 3 you should also indicate the BR17 points on there – maybe you could use crosses in the middle of the colours or something? Or maybe provide them all in separate plots in the Supplementary?

It would be useful to test how removing random data points affects the r^2 to get an idea of whether some of the results are due to chance rather than the particular sampling strategy.

Why just use r^2? Metrics other than r^2 should also be provided - e.g. mean bias, RMSE, etc. For the E-PEACE results in Fig. 3 for example the points removed by G18 sampling may make the r^2 worse (although perhaps due to the whole cluster at the top left being removed), but are likely to reduce the bias.

Table 2 – it would be useful to provide the number of samples in each case.

Compared to the in-situ comparison studies of Painemal (2011) and Kang (2021) you have many more datapoints for your comparison. Presumably this is because they selected entire profiles from the aircraft data and then found the 5x5 satellite pixels that were collocated (within an hour), but with a calculation to account for the movement of the cloud (using the wind speed and direction). It would be useful to comment on this in the paper. Those studies seemed to find much better r^2 values than you. Use of individual MODIS pixels also seems to lead to repeated sampling of the same regions of clouds (looking at Fig. 3), which may skew the statistics somewhat. Perhaps it would be better to average over a few pixels, e.g. 5x5 as in Painemal and Kang to see how this changes the results? Especially since the satellite and aircraft will not be precisely collocated.

Is there an effect from not correcting for the wind speed and direction between the aircraft and satellite observations as done in previous studies?

The results in Fig. 7 are a bit strange given the results from the G18 review where the BR17 retrievals (although without the G18 sampling applied here) tended to be lower than those from the other dataset tested (based on Grosvenor and Wood, 2014). Can these results be explained?

**Line by line comments**

p.1 L13 – "As the first moment of the droplet size distribution, the $N_d$ is important for setting cloud and precipitation process rates"

      Nd is the zeroth moment, not the first. Plus the fact that it is the zeroth moment doesn't seem that directly relevant for setting cloud and precipitation rates – perhaps it would clearer/relevant to say that Nd helps to determine the droplet sizes (or something similar)?

P1, L18 – you could also add https://doi.org/10.1029/2020MS002126 and https://doi.org/10.5194/acp-20-15681-2020 as examples of GCMs being evaluated using Nd.

p.2, L6 – Is Hasekamp available yet?

P3, L12 – "but the temperature dependence can produce a 50% variation in the Nd"

      It would be better to more specific here – is this for temperatures typical of the range of cloud temperatures encountered? All clouds, or just shallow ones, etc.

P4, L1 – "and a degeneracy in the retrievals for a low re"

      Needs some explanation of what this means in this context.

P4, L5 – "generate uncertainties, particularly in the re" – the Grosvenor paper actually showed larger optical depth effects than re at high solar zenith angles.

P7, L15 – can you provide more details on the particular coefficient of determination metric used. Is it the square of the correlation coefficient (which method)? Or is some other metric used?

Table 1 – this lists the 5km CF > 0.9 as a sampling criteria for G18, but the text suggests that this is not the case?

Fig. 3 – the green and orange dot colours don't stand out as being different enough from each other. Can you change the colour something else? Black would work well I think.

Eqn. 3 – can you provide a derivation of this please, or a reference? Using the definition of beta in Eqn.2 seems to require dividing by A when I derive it. Or using $\Delta \ln A$ instead of $\Delta A$. Does this affect the results?

Fig. 4b – it would be more useful if you tested the G18, BR17 and Z18 sampling without the SPI<30 restriction to see the variation across the full range.

Fig. 5 – why the sudden switch to using 1.6um retrievals?

P14, L4 – "In these regions, clouds are much more likely to be adiabatic (and hence satisfy the BR17 re stacking criterion)."

The 2.1 and 3.7um retrievals actually are usually quite close to each other in stratocumulus regions (e.g., see Fig. 1 of Painemal 2011), or if anything re2.1 > re3.7 and yet there is a general good match between in-situ and satellite Nd. This raises some issues with the use of the stacking sampling.

**Typos**
"Insitu" – should be two separate words "*in situ*" usually in italics.

---

## Author Comment (AC1)

**Reviewer 1**

*: This paper compares MODIS droplet concentration retrievals with those from several aircraft campaigns and calculates r2 values for several different sampling strategies for the MODIS pixels. The quality of the writing is good and a good number of field campaigns are included. The field was in need of such comparisons between in situ and satellite Nd data since previously this had only been done for a few campaigns. The overall scientific quality is good and lots of interesting aspects are explored. However, there are a few mistakes (concerning the values used for filtering relative to those suggested in the cited literature) and an over reliance on the r2 metric, which is not always informative. Additional metrics (mean bias, RMSE, etc.) should be calculated too. The aircraft sampling method is also quite different to previous comparisons leading to more datapoints but without the ability to determine the root cause of any discrepancies (since profiles are not used). This is fine, but it would be good to explain the differences with previous work (where higher r2 values were found) and to compare the effects of averaging larger numbers of pixels (e.g., 3x3, 5x5) rather than jumping to 100x100. I would also like to see uncertainties for the r2 values since it is unclear whether some of the differences in r2 are significant and there is likely a degree of randomness.*

**Reply**: We thank the reviewer for their useful comments. These are addressed below. The difference in the evaluation method is intentional, with the aim being to compare the $N_d$ retrieval to aircraft data under real-world conditions. This means that there is no extra data available, other than that from the satellite. While filtering for adiabatic profiles would increase the $r^2$ (in line with previous studies), if the satellite is unable to perform this filtering, this data would be included in an $N_d$ climatology based on the satellite data.

Even under these arduous conditions, the satellite retrieval performs remarkably well, with high $r^2$ values particularly in liquid stratocumulus campaigns. This is also the driver behind the jump to 100km as an averaging distance, as 1 by 1 degree is a commonly used resolution in satellite-based studies. The text has been modified to make this clearer, particularly in the conclusions where it is now noted that these $r^2$ values are smaller than in previous studies.
* * *
***Here is a list giving more detail on the above and some additional comments :-:*** *Grosvenor (2018) suggested that solar zenith angle was restricted to below 65 degrees not 55. And it was 55 degrees for the viewing angle, not 41.4. This will lead to quite a lot more data being discarded than would be the case if using the correct values. What effect does this have?*

**Reply**: Thank you for these excellent points. While text had stated values more in line with the earlier Grosvenor and Wood (2014) paper, the analysis was actually using values from the Grosvenor et al. (2018) review paper. The text has been modified to include the correct values.
* * *
*: For the G18 sampling there are lots of factors being applied at once – it would be better to test them individually. Which has the largest influence? It would*

*also be useful to test BR17 and Z18 separately from G18.*

**Reply**: This is an interesting question. While there are many factors that could be affecting the accuracy of the $N_d$ retrieval, testing them all is out of scope for this initial work. Future work is planned developing a more accurate $N_d$ retrieval by better identifying cases where the current ones fail. BR17 and Z18 are calculated on top of G18 are they consider different physical attributes of the clouds and the retrieval. While G18 is primarily concerned with the accuracy of the components of the retrieval, BR17 and Z18 make statements about the properties of the clouds. For this reason, we considered it a better comparison to apply them on top of G18, as they would also benefit from the selection of more accurate retrievals. The conclusions have been modified to highlight this and the methods have been re-worded to make this distinction clearer.
* * *
*: Can you provide uncertainties for the $r^2$ values? These would help determine the likelihood of the differences between the different sampling strategies being due to chance.*

**Reply**: The 5 and 95% bounds for the "all data points" conditions are now included as rows in Tab. 2.
* * *
*: Is $r^2$ the best metric to use? And you should provide more details on the particular $r^2$ value that you are using – is it appropriate for the data populations? Some of the $r^2$ values don't seem to match with what might be expected in Figs. 3 and 5 (admittedly judging by eye). E.g., are the COPE $r^2$ values really so low? Can you double check? Adding the uncertainties would help here. It would be helpful to also plot the lines of best fit against which the $r^2$ values are calculated rather than just the 1:1 line. For Fig. 3 you should also indicate the BR17 points on there – maybe you could use crosses in the middle of the colours or something? Or maybe provide them all in separate plots in the Supplementary?*

**Reply**: As the reviewer notes, one metric is not enough to characterise the distribution. We focussed on the $r^2$ as it is relatively well correlated to the RMSD and bias (two other important measures of the accuracy of the retrieval). The RMSD and mean bias for all campaigns together are now included in Tab. 2 and discussed in the text.

The COPE $r^2$ values are indeed low, this is likely due to the very difficult situation for the $N_d$ retrieval, with large amounts of non-adiabatic cloud.
* * *
*: It would be useful to test how removing random data points affects the $r^2$ to get an idea of whether some of the results are due to chance rather than the particular sampling strategy.*

**Reply**: We have now applied a bootstrap to generate the 5 and 95% bounds for the $r^2$, with values given in Tab. 2
* * *
*: Why just use $r^2$? Metrics other than $r^2$ should also be provided - e.g. mean bias, RMSE, etc. For the E-PEACE results in Fig. 3 for example the points removed by G18 sampling may make the $r^2$ worse (although perhaps due to the whole cluster at the top left being removed), but are likely to reduce the bias.*

**Reply**: Mean bias and RMSD for the entire dataset are now included in Tab. 2. The full tables are included in a supplement.
* * *
*: Table 2 – it would be useful to provide the number of samples in each case. Compared to the in-situ comparison studies of Painemal (2011) and Kang (2021) you have many more datapoints for your comparison. Presumably this is because they selected entire profiles from the aircraft data and then found the 5x5 satellite pixels that were collocated (within an hour), but with a calculation to account for the movement of the cloud (using the wind speed and direction). It would be useful to comment on this in the paper. Those studies seemed to find much better $r^2$ values than you. Use of individual MODIS pixels also seems to lead to repeated sampling of the same regions of clouds (looking at Fig. 3), which may skew the statistics somewhat. Perhaps it would be better to average over a few pixels, e.g. 5x5 as in Painemal and Kang to see how this changes the results? Especially since the satellite and aircraft will not be precisely collocated. Is there an effect from not correcting for the wind speed and direction between the aircraft and satellite observations as done in previous studies?*
**Reply**: The number of samples are shown in Fig. 6, this is now noted in the caption for Fig. 2.

As the reviewer notes, previous studies have conducted careful comparisons, using profiles of $r_e$ to identify adiabatic cases, providing the best possible situation for the satellite retrieval and diagnosing errors. In this study, we take a different approach, using only the satellite to filter the pixels for the comparison (other than ensuring the aircraft data is located in regions with little mixing). This provides a test for the retrieval close to how it is used in many aerosol-cloud studies.

A wind and parallax correction are now applied to the study here. They have a small impact on the results as they typically result in only a few pixel movement (and the cloud fields are largely similar at these scales).
* * *
*: The results in Fig. 7 are a bit strange given the results from the G18 review where the BR17 retrievals (although without the G18 sampling applied here) tended to be lower than those from the other dataset tested (based on Grosvenor and Wood, 2014). Can these results be explained?*
**Reply**: This is likely due to the BR17 sampling being applied on top of the G18 conditions. This is now given more prominence in the methods and conclusions.

**Line by line comments**
* * *
*p.1 L13*: *"As the first moment of the droplet size distribution, the Nd is important for setting cloud and precipitation process rates" Nd is the zeroth moment, not the first. Plus the fact that it is the zeroth moment doesn't seem that directly relevant for setting cloud and precipitation rates – perhaps it would clearer/relevant to say that Nd helps to determine the droplet sizes (or something similar)?*
**Reply**: The first clause of this sentence has been removed

***P1, L18****: you could also add https://doi.org/10.1029/2020MS002126 and https://doi.org/10.5194/acp-20-15681-2020 as examples of GCMs being evaluated using Nd.*
**Reply**: Thank you for the suggestions, they are added here
* * *
***p.2, L6****: Is Hasekamp available yet?*
**Reply**: Amended
* * *
***P3, L12****: "but the temperature dependence can produce a 50% variation in the Nd" It would be better to more specific here – is this for temperatures typical of the range of cloud temperatures encountered? All clouds, or just shallow ones, etc.*
**Reply**: Changed to "Assuming a saturated adiabatic lapse rate, the pressure dependence is weak, but a temperature change from 270K to 300K can double the condensation rate and hence the $N_d$."
* * *
***P4, L1****: "and a degeneracy in the retrievals for a low re" Needs some explanation of what this means in this context.*
**Reply**: Amended to "... a degeneracy in the retrievals for a low $r_e$ (where multiple $\tau_c$, $r_e$ combinations have the same reflected radiances), ..."
* * *
***P4, L5****: "generate uncertainties, particularly in the re" – the Grosvenor paper actually showed larger optical depth effects than re at high solar zenith angles.*
**Reply**: Good point. We have changed this sentence to "Maddux et al. (2010) and Grosvenor and Wood (2014) demonstrated the uncertainties at high solar zenith and satellite viewing angles, where cloud 3D effects and multiple scattering generate uncertainties, in both $r_e$ and $\tau_c$."
* * *
***P7, L15****: can you provide more details on the particular coefficient of determination metric used. Is it the square of the correlation coefficient (which method)? Or is some other metric used?*
**Reply**: Changed to " Coefficients of determination ($r^2$ - the square of the Pearson product-moment correlation coefficient), ..."
* * *
**Table 1**: *this lists the 5km CF ¿ 0.9 as a sampling criteria for G18, but the text suggests that this is not the case?*
**Reply**: A new sentence is added to the methods " To select more homogeneous cloud cases, pixels with a 5km cloud fraction less than 0.9 are also excluded."
* * *
**Fig. 3**: *the green and orange dot colours don't stand out as being different enough from each other. Can you change the colour something else? Black would work well I think.*
**Reply**: The green dots are now black.
* * *
**Eqn. 3**: *can you provide a derivation of this please, or a reference? Using the definition of beta in Eqn.2 seems to require dividing by A when I derive it. Or using $\Delta lnA$ instead of $\Delta A$. Does this affect the results?*
**Reply**: Thank you for spotting this. The equation has now been amended

to correctly use logarithms, in line with previous work. As a constant factor, it doesn't have a large impact on the forcing scaling between the sampling methods.
* * *
*Fig. 4b: it would be more useful if you tested the G18, BR17 and Z18 sampling without the SPI¡30 restriction to see the variation across the full range.*
**Reply**: We agree that to would be interesting to see how different factors affect the accuracy of the $N_d$ retrieval. This is a topic for future work though and we prefer to limit this paper to only potential improvements in the sampling strategies listed.
* * *
*Fig. 5: why the sudden switch to using 1.6um retrievals?*
**Reply**: This is because (strangely), the 1.6 µm retrieval appears to have the best correlation with insitu measurements at these larger scales. We are not clear exactly why this is, but speculate that it might be due to cloud top entrainment mixing. Further work is required to examine this possibility before a clear recommendation can be made here. The caption and conclusions have been modified to make this clearer.
* * *
*Fig 5.: A quick follow on from my review - can you specify for Fig. 5 whether data from all campaigns was used? Also there is a typo in the caption: "leave" instead of "least".*
**Reply**: A new sentence is added to the text "While there is not an explicit selection for specific campaigns, these representation criteria implicitly bias the results in Fig. 5 towards the liquid stratocumulus campaigns."
* * *
*P14, L4: "In these regions, clouds are much more likely to be adiabatic (and hence satisfy the BR17 re stacking criterion)." The 2.1 and 3.7um retrievals actually are usually quite close to each other in stratocumulus regions (e.g., see Fig. 1 of Painemal 2011), or if anything re2.1 ¿ re3.7 and yet there is a general good match between in-situ and satellite Nd. This raises some issues with the use of the stacking sampling.*
**Reply**: This is a good point - the stacking criterion is by no means perfect. It is not clear that all clouds that satisfy the stacking criterion are necessarily adiabatic and retrieval biases and cloud top entrainment could also affect the size order, shifting it from the simple adiabatic model. The sentence has been modified to "In these regions, clouds are much more likely to be adiabatic (and so more likely to satisfy the BR17 $r_e$ stacking criterion)"
* * *
*: "Insitu" – should be two separate words "in situ" usually in italics*
**Reply**: Amended, although the journal style does not have in situ in italics.

**Reviewer 2**
* * *
*: This manuscript compares data sampling strategies employed in several studies of cloud droplet number concentration (Nd) retrievals from MODIS observations. It also compares retrieved values using these strategies with in situ*

*observations from several aircraft campaigns. Accurately estimating Nd from satellite has been a challenging issue, given the number of artifact sources encountered in the remote sensing of cloud properties in the visible and near IR spectrum. An intercomparison of these sampling strategies is of benefit to those involved in cloud remote sensing as well as those validating models with MODIS estimates of Nd.*
* * *
*: I found the science to be sound and the manuscript well written. However, I was disappointed that the study was largely limited to the r2 metric when comparing strategies in the absence of in situ observations. While this is a good first order metric in assessing agreement in a relationship, I think including a comparison of measurement bias between techniques or comparing them to a retrieval that is agnostic with regard to filtering technique would improve the manuscript. Additionally, I think it would be beneficial to include error estimates, especially if considering bias. This could be as simple as propagating forward the uncertainties of optical thickness and effective radius given in the MODIS cloud product and reasonable assumptions on systematic or random errors of the other input parameters.*

**Reply**: We thank the reviewer for their useful comments. Following this comment and that from the other reviewer, we have included information on the bias and RMSD in Tab. 2, Fig. 4 and supplementary information. In most cases, these are correlated to the $r^2$, but as the reviewer notes, it is good to include them, particularly as the original point of this paper is that some sampling methods may generate biases. As we are using the same retrieval as described in Grosvenor et al. (2018), we do not reproduce their analysis here. However, we now note in the discussion/conclusions section that the normalised RSMD is smaller than that from Grosvenor et al. (2018), due in part to the larger number of points from stratocumulus regimes and partly due to the success of the sampling strategies.

**Individual comments:**
* * *
**Throughout the manuscript**: *Please change "insitu" to "in situ" and italicize.*

**Reply**: Amended
* * *
**P4L1**: *Using "degeneracy" will likely confuse readers that don't have a physics background. Perhaps refer to it is "ambiguous retrievals for small re"?*

**Reply**: This sentence has been modified to "...a degeneracy in the retrievals for a low $r_e$ (where multiple $\tau_c$, $r_e$ combinations have the same reflected radiances), ..."
* * *
**P11L1**: *While 15 microns is an upper limit associated with drizzle above that threshold, would using something like the H3/N ratio in vanZanten, et al. (2005) result in the rejection of less data greater than 15 microns while preserving the assumption of an adiabatic profile? Since geometrical thickness is relatively simple property to calculate with N, it could easily be used to filter data on the*

*fly.*

**Reply**: Given the regular use of 15 μm to indicate drizzle, we have included in here as an example of a filter that might be used in the future (and has been implicitly used in the past; Rosenfeld et al., 2019). We prefer to keep the 15 μm threshold for the simplicity of the interpretation, but have included a reference to vanZanten et al. (2005) and the more sophisticated measure of drizzle will aid future work in this area developing a more reliable $N_d$ retrieval.
* * *
**P11 Table 3**: *Please place the wavelengths in the table in ascending order.*
**Reply**: Amended
* * *
**P11L10**: *For estimating Nd, wouldn't 3.7 microns generally be the preferred wavelength since the re used in Eq 1 is assumed to be at cloud top? I think section 3.2.4 could be eliminated since regardless of r2, a retrieval from 1.6 microns is generally of limited utility.*
**Reply**: We agree and had previously assumed that the 3.7 μm retrieval would perform the best, given previous studies have shown it to be more accurate in broken-cloud conditions and closer to the top of the cloud.

The importance of this section is that is apparently not the case. At larger spatial scales in particular, the 1.6 μm retrieval appears to have a better correlation to the in situ data (Fig. 5). One possibility is that the 1.6 μm retrieval is less sensitive to cloud top mixing (being theoretically located deeper in the cloud), but further studies are necessary to precisely determine the cause of this effect. We note in the recommendations that the 1.6 μm retrieval should not yet be preferred because of this.
* * *
**Figure 1**: *Please add a legend to this figure.*
**Reply**: Amended
* * *
**Figure 5**: *Why is 1.6 microns being used here. Wouldn't 3.7 or 2.1 be a better choice for comparison?*
**Reply**: As above, the 1.6 μm retrieval is used here because it, surprisingly, has the best correlation to the insitu data at a large scale. The reasons for this are not yet clear (and this is now more clearly noted in the results and conclusions).

**Bibliography**

Grosvenor, D. P. and Wood, R.: The effect of solar zenith angle on MODIS cloud optical and microphysical retrievals within marine liquid water clouds, Atmos. Chem. Phys., 14, 7291–7321, https://doi.org/10.5194/acp-14-7291-2014, 2014.

Grosvenor, D. P., Sourdeval, O., Zuidema, P., Ackerman, A., Alexandrov, M. D., Bennartz, R., Boers, R., Cairns, B., Chiu, J. C., Christensen, M., Deneke, H., Diamond, M., Feingold, G., Fridlind, A., Hünerbein, A., Knist, C., Kollias, P., Marshak, A., McCoy, D., Merk, D., Painemal, D., Rausch, J., Rosenfeld, D., Russchenberg, H., Seifert, P., Sinclair, K., Stier, P., van Diedenhoven, B., Wendisch, M., Werner, F., Wood, R., Zhang, Z., and Quaas, J.: Remote Sensing of Droplet Number Concentration in Warm Clouds: A Review of the Current State of Knowledge and Perspectives, Rev. Geophys., https://doi.org/10.1029/2017RG000593, 2018.

Maddux, B. C., Ackerman, S. A., and Platnick, S.: Viewing Geometry Dependencies in MODIS Cloud Products, J. Atmos. Ocean. Tech., 27, 1519–1528, https://doi.org/10.1175/2010JTECHA1432.1, 2010.

Rosenfeld, D., Zhu, Y., Wang, M., Zheng, Y., Goren, T., and Yu, S.: Aerosol-driven droplet concentrations dominate coverage and water of oceanic low level clouds, Science, p. eaav0566, https://doi.org/10.1126/science.aav0566, 2019.

vanZanten, M. C., Stevens, B., Vali, G., and Lenschow, D. H.: Observations of the Structure of Heavily Precipitating Marine Stratocumulus, J. Atmos. Sci., 62, 4327–4342, https://doi.org/10.1175/JAS3611.1, 2005.

---

## Referee Report (RR1)

**Review of Gryspeerdt, et al. - The impact of sampling strategy on the cloud droplet number concentration estimated from satellite data**

I'm satisfied with the responses to the comments and the new manuscript except for a few small things as noted below.

I think that Eqn.3 is still wrong – it needs to be multiplied by $\alpha_c$. Presumably this does not affect the results.

Line numbers refer to the tracked changes version of the new manuscript here.

**Typos**

L105 – Should be "while BR17 and Z18" rather than "which"?

L216 – "as the minimum cloud fraction…."?

L256 – Capital letter before "the 2.1um" and "which" should be "while".

L406- "G18 has the closest match to the climatology (Fig. 6)," – not sure if I can see this from the plot? A panel for all of the campaigns combined would be useful.

---

## Author Response (AR2)

**Reviewer 1**

We thank the reviewer for their careful reading of the revisions and have incorporated all of their changes. Additional text has been added describing the new panel in Fig. 6 (the weighted average climatology of all the campaigns).

"The distribution for the complete dataset is dominated by the stratocumulus campaigns, particularly E-PEACE. The similarity of the $N_d$ from the different sampling strategies (Fig. 7) means that there is relatively little variation between the regimes, although BR17 and Z18 (and to a lesser extent G18) have a narrower $N_d$ range compared to the *in situ* data. Weighting each campaign equally (Fig. 6l) shows that for these campaigns, BR17 has the tendency to remove the lowest $N_d$ values (giving it a slightly high bias) and Z18 to remove the highest."

While amending equation 3, we found that not all of the years of data were being included in the calculation of the forcing enhancement (Fig. 9). This makes a small change to magnitude of the forcing enhancement estimates (lower right corners of the figures), but does not change the interpretation of the results.

Additional small changes have been made to fix typos around lines 365 and 400 in the diff.